# Classification of Events Violating the Safety of Physical Layers in Fiber-Optic Network Infrastructures

**DOI:** 10.3390/s22239515

**Published:** 2022-12-06

**Authors:** Michal Ruzicka, Lukas Jabloncik, Petr Dejdar, Adrian Tomasov, Vladimir Spurny, Petr Munster

**Affiliations:** Department of Telecommunications, Faculty of Electrical Engineering and Communications, Brno University of Technology, Technicka 12, 616 00 Brno, Czech Republic

**Keywords:** event classification, machine learning, optical fiber sensor, physical layer security, state of polarization changes, vibration

## Abstract

Fiber-optic network infrastructures are crucial for the transmission of data over long and short distances. Fiber optics are also preferred for the infrastructure of in-building data communications. In this study, we use polarization analysis to ensure the security of the optical fiber/cables of the physical layer. This method exploits the changes induced by mechanical vibrations to polarization states, which can be easily detected using a polarization beam splitter and a balancing photodetector. We use machine learning to classify selected events that violate the safety of the physical layer, such as manipulation or temporary disconnection of connectors. The results show the resting state can be accurately distinguished from selected security breaches for a fiber route subjected to environmental disturbances, where individual events can be classified with nearly 99% accuracy.

## 1. Introduction

Most optical sensor systems are based on reflectometry or interferometry principles and are highly sensitive and accurate, but also extremely expensive. Systems based on commercial polarimeters can accurately measure polarization in optical fibers, but are costly and mostly have a low sampling rate [1]. Detection systems based on a simple analysis of changes in polarization states are a cost-effective solution that can be easily implemented for current security systems. Problems with data processing are also encountered using optical sensor systems. Algorithms have been developed that can increase the signal-to-noise ratio [2,3], filter signals [4,5], or amplify signals [6]. A threshold is then used mainly to determine the increase in the vibration intensity, which is suitable only for a certain type of event that occurs close to the optical fiber. The measurement results of polarization changes are therefore highly inaccurate. The measurement accuracy can be increased by using a simple polarization analyzer and a data acquisition device in conjunction with machine learning algorithms to identify changes in polarization states. It is important to process the data acquired from the polarization analyzer by using advanced event classification techniques rather than thresholding. The most suitable classification techniques involve the use of machine learning algorithms, which can also determine the type of event that occurs in the vicinity of a measured optical fiber.

## 2. Related Studies

Several technologies can be used to protect the physical layer consisting of optical fibers/cables. In [7], it was noted that most common attacks on the physical layer of fibers involve touching the fiber. Thus, vibration detection can be used to ensure the security of the physical layer consisting of optical fibers/cables. Special fibers, such as fiber Bragg gratings (FBG), can be used for vibration detection. These systems are very often used to provide perimeter security, such as in [8,9]. Vibrations in the secure zone can thus be detected, but unfortunately, the use of special fibers with Bragg gratings is necessary. Thus, FBG cannot be used in a real infrastructure.

Another potential technique for ensuring fiber security is distributed acoustic sensing (DAS) based on Φ-OTDR. These systems are very accurate and enable localization of events and measurements over long distances, while being complex, robust, and costly [3,10]. Consequently, these systems are often used to secure critical infrastructures [11,12].

Another option is to measure vibrations using interferometers. The most commonly used techniques employ the Mach–Zehnder interferometer (MZI), which can be used, for example, to secure gas turbines [13]; the Michelson interferometer (MI), which is commonly used for perimeter security [14]; or the Sagnac interferometer (SI), which is also used for perimeter security [15,16]. The use of the MZI and MI requires the use of reference fibers [17], which can be complex to implement in real infrastructures. Interferometric methods are very sensitive and hence can be disturbed by common traffic sounds near real cables in buildings (banging doors; noise from heating, ventilation, and air conditioning (HVAC); people moving in buildings, etc.).

An example of the enhanced sensitivity of interferometers is provided in [18] by comparison to the sensitivity of polarization detection, which constitutes another large group of sensing techniques based on the detection of polarization changes in a fiber. For example, polarization analysis can be performed using polarization-sensitive optical time domain reflectometry (POTDR) based on Rayleigh light scattering [19]. Another possibility is to use a polarimeter to measure the polarization of the light and use machine learning to detect polarization changes [20] or to use a polarization beam splitter (PBS) and a balanced photodetector (BPD) to directly measure polarization changes [21,22]. Measurements using a PBS can also be incorporated into reflectometry methods [23], that is, PBS can be used for POTDR sensing.

Sensing systems based on polarization detection appear to be an ideal solution because of low acquisition costs. These systems could be easily integrated into real infrastructures, either by employing unused fibers in cables or wavelength-division multiplexing in fiber networks with active data traffic. Paradoxically, these systems are less sensitive than interferometers and are therefore more robust to real-world conditions.

It has been proven in many research studies that optical networks are vulnerable to eavesdropping attacks [7,24,25]. However, various fiber optics sensing systems can detect or even localize the attacker. These methods can be extended by using machine learning algorithms to improve attacker detection. One example is a detection-and-localization system based on optical time domain reflectometry (OTDR) combined with a convolutional neural network [26] or a recurrent autoencoder neural network [27]. These methods achieved almost 100% accuracy in attacker detection and localization, demonstrating the efficacy of such a solution. A different approach is to focus on attacker detection without localization to reduce the size and price of the proposed solution. This kind of system can analyze optical side-channel information, such as transmission parameters obtained from eye diagrams using convolutional neural networks [28], or analyze polarization changes using simple threshold algorithms and data processing [29].

## 3. Methodology

The propagation of light can be described as a transverse electromagnetic wave with perpendicular electric and magnetic components. The propagation of these components can be described by Maxwell’s equations. Such a wave can be described as a superposition of two orthogonal and linearly polarized components along the x- and y-axes, where the respective amplitudes and initial phases are given by [30]
(1)$$\begin{array}{cc} \; & E_{x}=a_{x}\cos\left(\omega t-kz+\delta_{x}\right),\\ \; & E_{y}=a_{y}\cos\left(\omega t-kz+\delta_{y}\right), \end{array}$$
where $k=(2\pi /\lambda_{0})n$, *n* is the refractive index, $\lambda_{0}$ is the wavelength, *z* is the direction of propagation, and *a* and $\delta_{0}$ are the amplitude and the initial phase, respectively [30].

The graphic progression is shown in Figure 1 [17].

Equation (Equation 1) can then be written using a Jones vector representation as a 2×1 matrix (column vector) [31]:(2)$$E\left(z,t\right)= [\begin{array}{c} E_{x}\\ E_{y} \end{array}]= [\begin{array}{c} a_{x}\cos\left(\omega t-kz+\delta_{x}\right)\\ a_{y}\cos\left(\omega t-kz+\delta_{y}\right) \end{array}].$$

The polarization of light can also be described by Stokes parameters, which are defined as
(3)$$\begin{array}{cc} \; & S_{0}=I_{0},\\ \; & S_{1}=I_{H}-I_{V},\\ \; & S_{2}=I_{+45}-I_{-45},\\ \; & S_{3}=I_{\mathrm{RCP}}-I_{\mathrm{LCP}}, \end{array}$$
where $I_{0}$ denotes the intensity of the light beam and $I_{H},I_{V},I_{+45},I_{-45},I_{\mathrm{RCP}}$ and $I_{\mathrm{LCP}}$ represent the transmitted intensities of a beam passing through a linear horizontal polarizer (LHP), a linear vertical polarizer (LHP), and a right-circular and left-circular polarizer, respectively.

Stokes parameters are very often represented by Stokes vectors represented by the 4×1 matrix [32]:(4)$$S= [\begin{array}{c} S_{0}\\ S_{1}\\ S_{2}\\ S_{3} \end{array}]\;\mathrm{or}\;\left\{S_{0},S_{1},S_{2},S_{3}\right\}.$$

Figure 2 shows a Poincaré sphere as a measure of the current state of polarization or the evolution of polarization over time. The V(S) and H(F) markings on the axes show the values of vertical and horizontal polarization on the fast or slow axis, respectively. The other axes represent the right- and left-circled axes [33].

The polarization can be defined in terms of three basic states, i.e., linear, circular and elliptical [30].

Linear—electric field concentrated in one plane along the direction of propagation.Circular—the electric field consists of two mutually perpendicular components with the same amplitude that are shifted by a phase $\frac{\pi }{2}$.Elliptic—The electric field can be described by an ellipse due to the different amplitudes and/or different phases of the components.

As mentioned in Section 2, vibrations around an optical fiber affect the polarization and can be measured in two ways. One option is to use a polarimeter to measure the current polarization state or to use a PBS to directly detect polarization changes. As we are designing our system to provide cost-effective security for a physical layer, it is more suitable and cheaper to use a polarization splitter to split the beam and then detect two separate orthogonal planes using a BPD [34,35].

We split the polarization into two planes and determine the signal intensity for both planes, where there are no changes at rest, i.e., the signal intensities in the two planes are probably different but constant. The action of an external force on the fiber changes the rotation of the polarization, thereby changing the ratio of the polarizations in the orthogonal planes. This change is detected by the BPD, which senses both planes and differentiates the signal. Even if there is a permanent change in the polarization, the proposed system will not be affected because the signal intensities from the PBS will also stabilize once the new state of polarization has stabilized. The resulting signal from the BPD is thus very similar to an audio mono signal, which would simplify conversion to this format.

## 4. Experimental Setup

The experimental setup used to test the proposed system consists of two parts. The first part is used for computation and consists of a computer and a device for sensing audio signals. The second part is an optical system with a laser source (a low-noise DFB laser with a 1550-nm wavelength and a typical spectral linewidth of 3 MHz), a PBS and a BPD (composed of InGaAs photodiodes with wavelengths of 900–1700 nm and a signal bandwidth of 100 MHz at 3 dB). The entire setup is located in a server room, from which an optical fiber leads out of the building within the test infrastructure. The whole design is shown in Figure 3, where the optical connections are marked in black, orange and blue. Signal and power connections are marked in red.

The computation section (marked in blue) originally consisted of a server and a myRIO DAQ board. However, the insufficiently low sampling rate of 8 bits/20 kHz resulted in the introduction of large errors into the measurements. To optimize the computing power and device size, the server was replaced by a microcomputer UP Board UP 4000 series with an Intel Pentium processor and 8 GB of RAM. The myRIO board was also replaced by an external sound card with a sampling rate of up to 24 bits/192 kHz. The final computational section is marked in green. The modified section ensures that the whole setup is almost pocket size.

A total of 3 ADCs were used for the measurements. A myRIO ADC board was used for signal conversion in the first version of the setup and was replaced in the second version by 2 external sound cards—Startech ICUSBAUDIO2D and Native Instruments Komplete Audio 2, with sampling rates of up to 96 kHz and 192 kHz, respectively. The most current version of the experimental setup is shown in Figure 4.

In [7], attacks on the physical layer are mainly considered to occur through tampering with an optical fiber. Therefore, measurement scenarios based on such situations were investigated. Measurements were performed under several scenarios. Specifically, direct manipulation of a fiber by touching and knocking, opening and closing of a manhole cover, disconnection of the fiber (which is particularly crucial for the safety of transmissions), and idle measurements were performed.

Measurements are performed on an optical fiber that passes through an entire building and then out around a road, as shown in Figure 5. The length of the optical route is approximately 2 km. There are many interference-producing elements along the fiber route. The measurement setup is located in a server room with a continuously operating HVAC. The fiber route passes above the ceiling along the corridor with a high turnover of people, continues to the lower floors of the building around a vertical riser, and then moves to an exterior manhole near the parking lot. The manhole is accessible to an operator or attacker, and the fiber is also accessible from the road along the fiber route to the server room. The proposed system targets direct manipulation in the vicinity of the cable/fiber by an attacker. The sensitivity of the system enables monitoring even in heavily disturbed surroundings and, therefore, in places where an interferometer could not be used because of high sensitivity.

## 5. Neural Network Architecture and Training

A neural network design consists of a simple architecture based on dense layers that can be easily offloaded into embedded systems for real-time analysis. The input layer has 8192 neurons (The window size is the optimal value considering the spectral resolution and computational difficulty.) to match the window-size generating spectrum vectors. The output layer matches the number of classes, which is equal to 5. The only optimized part is the middle layer. Based on the results of several experiments, the optimal number of neurons in this layer is 512. The activation function used between each layer is *sigmoid*, but the last layer uses a *softmax* function. The whole neural network and training are implemented using the PyTorch framework with graphics processing unit acceleration [36].

### 5.1. Dataset

The system described in Section 4 for analyzing changes in the polarization state is deployed to obtain a dataset. The system provides one-dimensional data in the time domain with a sampling rate of 44.1 kS/s that are not suitable for further analysis [29]. Therefore, the data are converted to the frequency domain using a single-side fast Fourier transform (FFT) and normalized by subtracting the mean value from each spectral vector.

Each abnormal event in the dataset is marked by starting and ending points, thereby generating arrays of different lengths for events. The data array for the FFT is generated by a sliding window with a size of 8192 elements that is shifted to 820 elements after each generated window. Thus, several spectral vectors are produced for each labeled abnormal event.

The dataset labels are converted into one-hot encoding, which a neural network can use to solve a classification problem.

The dataset is divided into three subsets in a ratio of 80:10:10 (This ratio is chosen to match the 10-fold cross-validation method described in Section 5.2.) as shown in Table 1. The training subset is used directly to modify the neural network weights, the validation subset only evaluates the loss of the neural network during learning to prevent overfitting, and the test subset is used only once after learning to evaluate the metrics and performance of the trained neural network.

Considering the problem to be solved and the neural network output, training is performed using the categorical cross-entropy loss function, which is defined as [37]
(5)$$\ell \left(y,\hat{y}\right)=-\sum_{i=1}^{N}y_{i}\log\hat{y_{i}}\;,$$
where *y* is a true label, $\hat{y}$ is a predicted output and *N* is the number of classes.

Table 1 clearly shows that the dataset is imbalanced. Therefore, the loss function is further enhanced using class weights to scale the output loss [38]. This modification decreases the losses for large classes and increases the losses for small classes.

### 5.2. Training

In the training process, 10-fold cross-validation is used to splits the dataset into 10 distinct subsets. Thus, the training is repeated 10 times, where the training, validation, and testing subsets vary according to the cross-validation method. In each iteration, the training process uses the Adam [39] gradient descent algorithm with a learning rate of $5\times 10^{-4}$ for 50 epochs. The number of epochs is evaluated in advance by using an early stopping mechanism to prevent overfitting. The learning process is terminated if the validation loss does not improve for more than 10 epochs. The proposed model contains 2,100,741 trainable parameters requiring approximately 5 seconds per epoch.

In Figure 6, the mean values of the accuracy and loss metrics for all the learning processes are shown by solid lines, and the 25th and 75th quantiles in each epoch are highlighted in the background. The training curves have low errors and converge to the lowest loss. By comparison, the validation curves, particularly the validation loss curve, have higher errors resulting from the use of the cross-entropy loss with class weights. A classification accuracy of almost 99% is achieved for both datasets. The performance evaluation for the test dataset is described in Section 6.

## 6. Results and Discussion

The measured results can be used to identify individual events. For example, fiber manipulation has a very significant effect on the acquired signal, as shown in Figure 7. This event can be observed in both the time domain (see Figure 7a) and the frequency domain (see Figure 7b), which is primarily used for machine learning and dataset labeling. The frequency domain is mainly used for labeling to improve visibility of events, such as knocking on a manhole. The visibility of these events need to be improved because the events only affect the polarization slightly.

As signal changes are visible on all acquisition devices, we performed an experiment to qualitatively assess the accuracy of the proposed method.

To compare the ADCs, a method was used to calculate the SNR in the frequency domain (i.e., the SNR was compared at each frequency in a predefined test signal, where sine functions were generated at a given frequency for 0.7 s, where the frequency increased in 1/12-octave steps). Figure 8 shows that the sound cards have a higher SNRfthan the myRIO board.

The parameters of each converter are presented in Table 2. Most sound cards use delta–sigma converters, whereas myRIO integrates an XADC directly into a Zynq 7010 processor. Thus, myRIO has a lower SNRf and bit resolution than sound cards.

We can easily compute the SNRf for a specific frequency *f* as
(6)$$\mathrm{SNRf}\left(f\right)=10\log_{10}\frac{\sum_{m=1}^{M}{\left|{\hat{x}}_{m}\right|}^{2}}{\sum_{m=1}^{M}{\left|x_{m}-{\hat{x}}_{m}\right|}^{2}},$$
where ${\hat{x}}_{m}$ is the sine signal with a given frequency and $x_{m}$ is the acquired signal. The value of the SNRf is expressed in decibels. Thus, it can be concluded that a sound card is more suitable for acquiring data than myRIO.

The aforementioned conclusion is confirmed by the spectrogram of the acquired signals shown in Figure 9. At first glance, the spectrum of the signal acquired by myRIO only contains frequencies up to 2 kHz, whereas higher frequencies appear in the spectrum of the signal obtained using the sound card. Both spectrograms contain strongly colored low frequencies. In myRIO, the higher frequencies are captured with very weak intensities or not at all, whereas higher frequencies are more prominent in the spectrum obtained using the Startech sound card. For this reason, we used the sound card data for machine learning. The steps at higher frequencies are generally clearly visible because the spectrum is computed using the partial frequencies in the test signal, i.e., equidistant frequencies are not used. This effect becomes more pronounced at higher frequencies because of the logarithmic nature of the steps.

The classification performance is estimated using a confusion matrix evaluated on the test dataset, which is shown in Figure 10. The proposed method is 100% accurate for deciding between the regular-state samples and the rest of the samples corresponding to abnormal events. Therefore, the network should not produce any false alarms. Satisfactory accuracy is obtained for identifying the remaining classes, except for the *Knocking* class, which is misclassified with *Manipulation*. An in-depth analysis shows that *Manipulation* events also contain knocking events, particularly at the beginning.

Unfortunately, the designed system cannot directly locate the origin of vibrations. It is possible to indirectly locate some types of vibration based on the known route along which the fiber is deployed. An example is the disconnection and possible reconnection of a fiber, which in our case can only occur in the server room, because there are no connectors elsewhere along the fiber route. Rack door disruption can also be determined by a class extension (e.g., the opening of a rack door in Server Room 1 and the opening of a door in Server Room 2) because server rooms frequently have different designs (e.g., sliding doors vs. opening doors).

The simultaneous classification of two different phenomena can also occur under real conditions, which we will investigate in the future. Primarily, we will extend the datasets while modifying machine learning algorithms, especially concerning frequency analysis, because events are frequency specific.

A unique feature of the proposed system is that polarization detection is used for protecting the physical layer and complemented by using machine learning algorithms for event classification, enabling the system to be easily deployed. The greatest advantages of the proposed system are a low cost, the ability to be deployed in real infrastructure, and compactness. The system can sense unused fiber in a cable to ensure the security of the entire cable. In the future, we would like to perform wavelet multiplexing or data traffic analysis.

## 7. Conclusions

In this article, the problem of vibration detection using optical fibers was described, and a complete system was designed and tested for detecting disturbances to a physical layer by measuring polarization changes using a PBS. The feasibility of securing a physical layer by a vibration detection system has been demonstrated, and events have been classified using a simple neural network based only on dense layers for potential attempts to violate the integrity of a physical layer consisting of optical fibers and cables. Experimental results prove that our proposed cost-effective system can prevent simple attempts at security breaches. The system can classify events with an accuracy of nearly 99%.

## Figures and Tables

**Figure 1 sensors-22-09515-f001:**
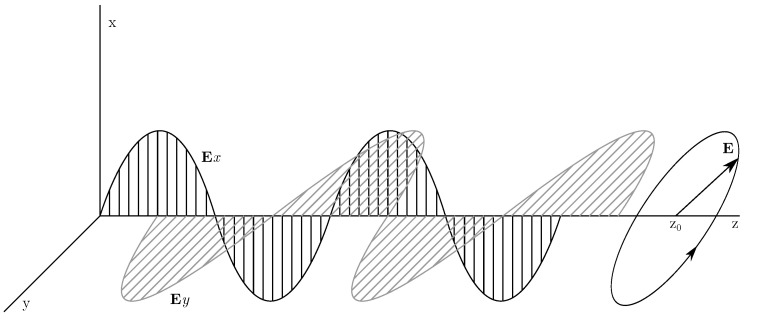
Polarization components of an electromagnetic wave **E**.

**Figure 2 sensors-22-09515-f002:**
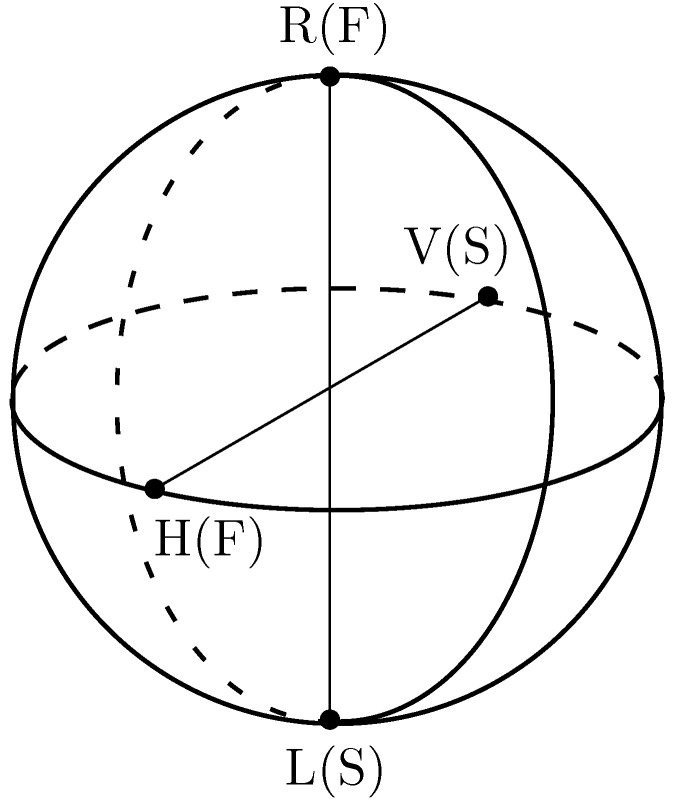
Poincaré sphere used to represent polarization rotation.

**Figure 3 sensors-22-09515-f003:**
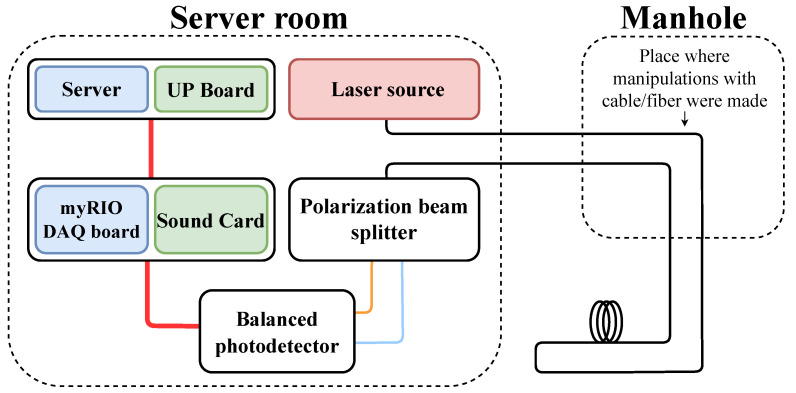
Schematic of the experimental setup.

**Figure 4 sensors-22-09515-f004:**
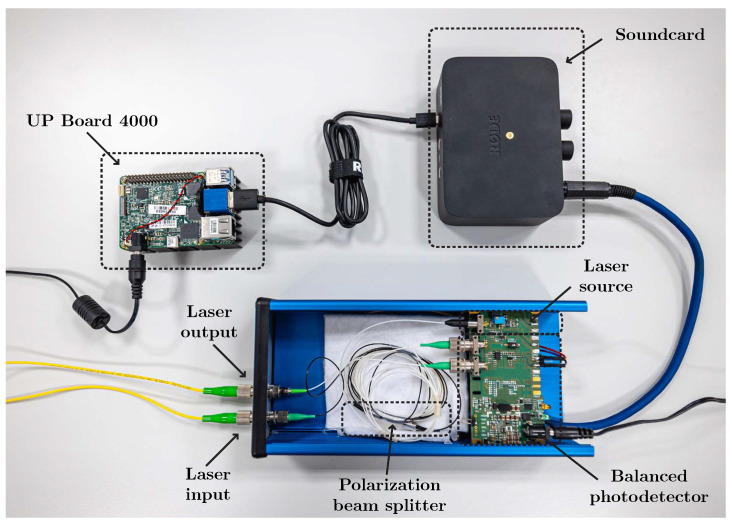
Final version of the sensor system with real hardware.

**Figure 5 sensors-22-09515-f005:**
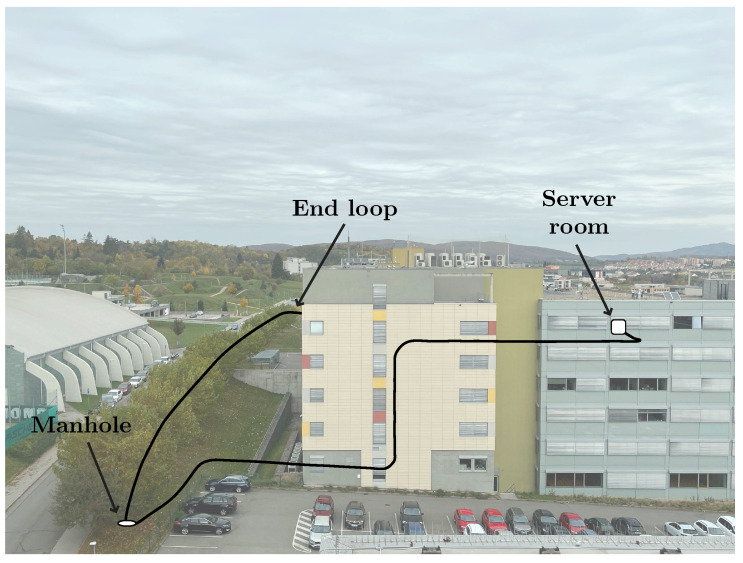
Drawing of a test route on a university polygon used to simulate a real infrastructure.

**Figure 6 sensors-22-09515-f006:**
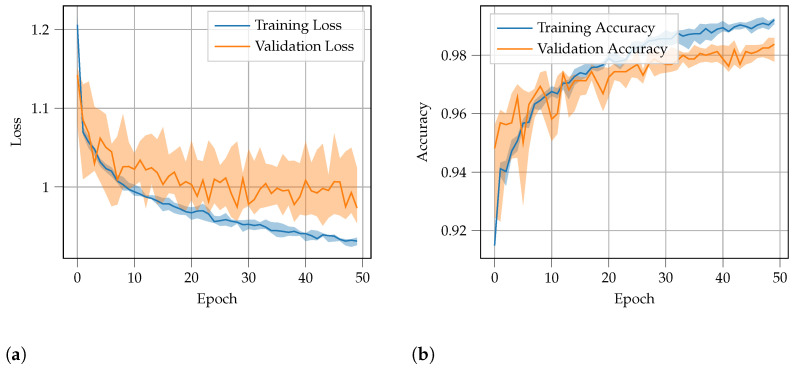
Evolution of the loss (**a**) and accuracy (**b**) over learning epochs with 10-fold cross-validation, where the line indicates the median values and the filled surroundings indicate the 25th and 75th quantiles for each epoch.

**Figure 7 sensors-22-09515-f007:**
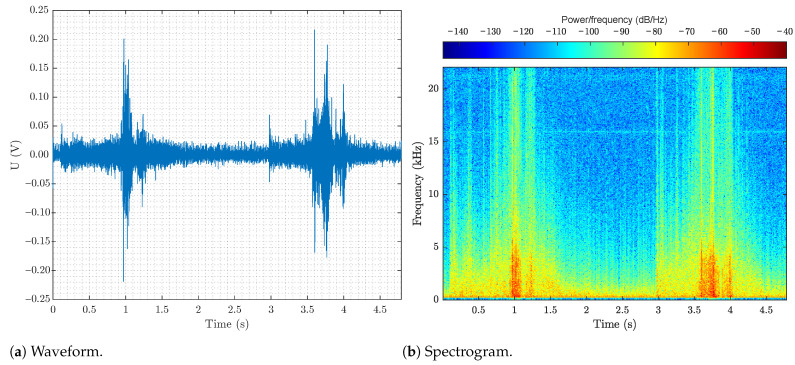
Example of a signal indicating two events of manipulation of fibers in a manhole.

**Figure 8 sensors-22-09515-f008:**
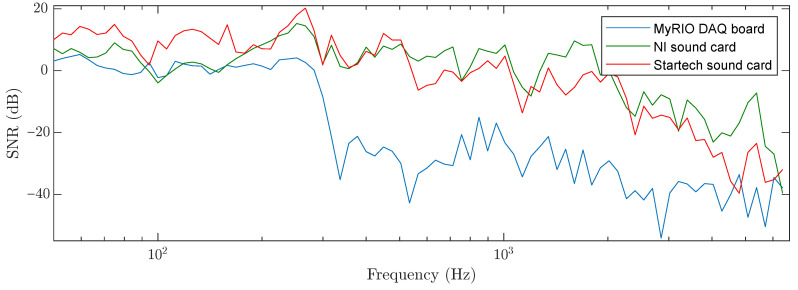
Comparison of the SNR in the frequency domain for myRIO and sound cards.

**Figure 9 sensors-22-09515-f009:**
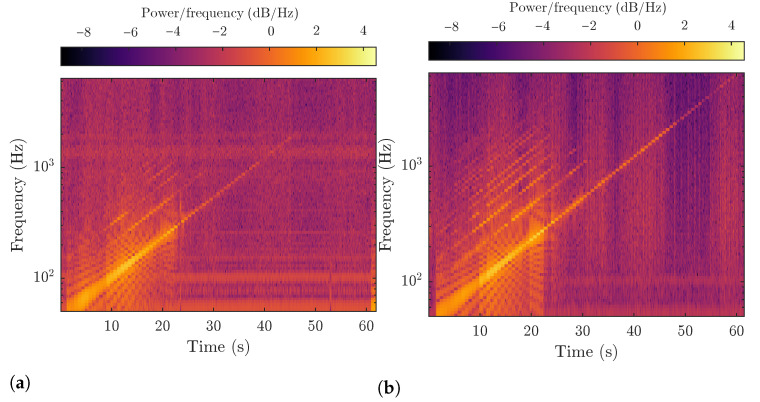
Spectrogram of test signals acquired by myRIO (**a**) and Startech (**b**).

**Figure 10 sensors-22-09515-f010:**
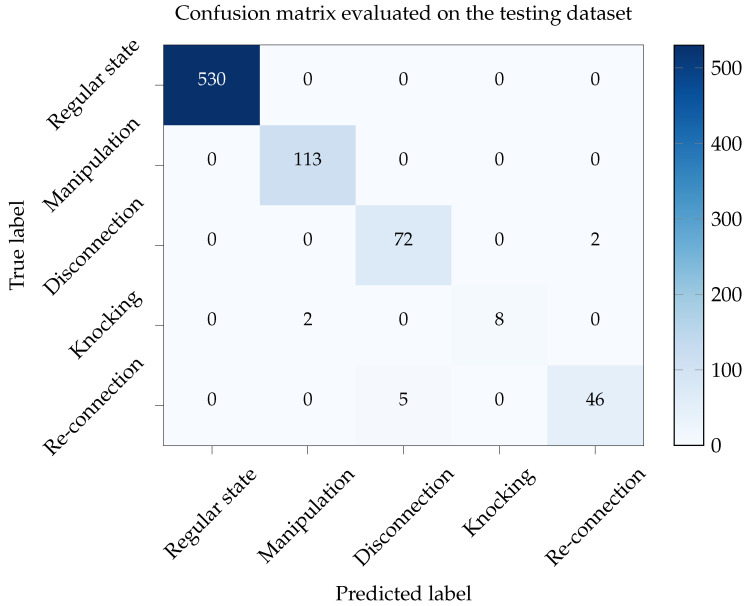
Confusion matrix evaluated on the testing dataset after learning, which shows the true performance of the proposed event classification method.

**Table 1 sensors-22-09515-t001:** The number of dataset samples per subset and label.

Label / Subset	Training	Validation	Testing	Total
Manipulation	918	115	115	1148
Physical disconnection	606	76	76	758
Re-connection	415	52	51	518
Knocking	93	10	10	113
Regular state	4247	531	530	5308
Total	6279	774	772	7845

**Table 2 sensors-22-09515-t002:** Parameters and prices of acquisition boards.

DAQ Device	ADC	Sample Rate	Bit Resolution	Price (EUR)
NI MyRIO 1950	XADC	up to 500 kHz	12	530
NI Komplete audio 2	delta-sigma	up to 192 kHz	24	129
Startech ICUSBAUDIO2D	delta-sigma	up to 96 kHz	24	43

## Data Availability

The data that support the findings of this study are available from the corresponding author upon reasonable request.

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
