# Peer review of "Classification of Events Violating the Safety of Physical Layers in Fiber-Optic Network Infrastructures"

_sensors, 2022, doi:10.3390/s22239515_

Round 1

Reviewer 1 Report

In this manuscript, the authors proposed polarization analysis to secure optical fiber/cables on the physical layer. The theoretical analysis and experimental results show that the proposed sensor is feasible. However, there are some points should be emphasized and interpreted. However, there are some points should be emphasized and interpreted.

1.The innovation points in the manuscript are not obvious, and the keywords are incomplete.

2. Pay attention to the capitalization of words in the manuscript. For example, “ Distributed Acoustic Sensing (DAS)” polarization-sensitive optical time domain reflectometry (POTDR), which should be consistent.

3.The characters in the formula should be given their meanings.

4. It is suggested to supplement the complexity or time of the algorithm.

5. Authors should carefully check the manuscript before submission.

Author Response

Response to the Report of Reviewer #1
We appreciate the effort of Reviewer #1 and we thank him/her for the comments and
suggestions.
In this manuscript, the authors proposed polarization analysis to secure optical fiber/cables on
the physical layer. The theoretical analysis and experimental results show that the proposed
sensor is feasible. However, there are some points should be emphasized and interpreted.
However, there are some points should be emphasized and interpreted.
We thank the Reviewer for feedback of our manuscript. We tried our best to properly treat
the issues mentioned below.
1. The innovation points in the manuscript are not obvious, and the keywords are incomplete.
We thank the reviewer for this comment, this is a very important part that has been over-
looked by us. Paragraph with innovations has been added to the Results and discussion
chapter.
2. Pay attention to the capitalization of words in the manuscript. For example, “ Distributed
Acoustic Sensing (DAS)” polarization-sensitive optical time domain reflectometry (POTDR),
which should be consistent.
We corrected the acronyms to be consintent.
3. The characters in the formula should be given their meanings.
We checked all the characters meanings in the formulas and they should be fine.
4. It is suggested to supplement the complexity or time of the algorithm.
We added a description of the complexity and time of the algorithm to the chapter Training.
5. Authors should carefully check the manuscript before submission.
We eliminated the deficiencies in the English language by professional proofreading.

Reviewer 2 Report

In my opinion this paper could be suitable for publication, however some issues must be addressed:

1. In the section Methodology, the authors only present the basic theory of Polarization. How is such methodology applied in the system?

2. What was the optical fiber test length?

3. It is necessary a description of characteristics of the optical part components

Author Response

Response to the Report of Reviewer #2
We appreciate the effort of Reviewer #2 and we thank him/her for the valuable comments, criticism and suggestions, which made the article clearer and more readable.
In my opinion this paper could be suitable for publication, however some issues must be addressed:
We thank the reviewer for positive feedback of our manuscript and recognition.
1. In the section Methodology, the authors only present the basic theory of Polarization.
How is such methodology applied in the system?
We apologize for the missing part. We added paragraph at the end of chapter Experimental setup.
2. What was the optical fiber test length?
We apologize for the missing fiber length. We added a information about the length to the chapter Experimental setup part about tested fiber.
3. It is necessary a description of characteristics of the optical part components.
We added a description of the optical components to the chapter Experimental setup.

Reviewer 3 Report

This manuscript proposes and verifies a physical layer disturbance detection system based on the detection of polarization changes using PBS. The results could be interesting for practical applications. However, there are some issues with the manuscript that should be addressed before publishing.

1.     Line 51, the abbreviation HVAC is not necessary, since it was not repeated in the manuscript.

2.     The caption of Figure 4 is not clear and should be detailed.

3.     The caption of Figure 5 is confusing and should be modified.

4.     In Figure 9, the high-frequency part of the signal becomes increasingly prominent. The authors should explain.

5.     In the manuscript, the optical system demonstrated the detection of a single disturbance, could the system recognize disturbances if two or more events occur simultaneously? In addition, could the system identify the location where the disturbance occurs? The authors should discuss.

Author Response

Response to the Report of Reviewer #3
We appreciate the effort of Reviewer #3 and we thank him/her for the comments and suggestions.
This manuscript proposes and verifies a physical layer disturbance detection system based on the detection of polarization changes using PBS. The results could be interesting for practical applications. However, there are some issues with the manuscript that should be addressed before publishing.
We thank the Reviewer for very positive feedback of our manuscript. We tried our best to properly treat the issues mentioned below.
1. Line 51, the abbreviation HVAC is not necessary, since it was not repeated in the manuscript.
We apologize but the reviewer seems to have overlooked. The abbreviation HVAC is repeated in chapter Experimental setup.
2. The caption of Figure 4 is not clear and should be detailed.
We modified caption of Figure 4 to be more detail.
3. The caption of Figure 5 is confusing and should be modified.
We apologize for confusion. We modify the caption of Figure 5 to a clearer form.
4. In Figure 9, the high-frequency part of the signal becomes increasingly prominent. The authors should explain.
We added a explanation of the spectrogram to the chapter Results and discussion.
5. In the manuscript, the optical system demonstrated the detection of a single disturbance, could the system recognize disturbances if two or more events occur simultaneously? In addition, could the system identify the location where the disturbance occurs? The authors
should discuss.
We apologize, this is really relevant comment. We added a two paragraphs about localization and combination of event to the chapter Results and discussion.
